# Pyruvate Kinase M2 Promotes Hair Regeneration by Connecting Metabolic and Wnt/β-Catenin Signaling

**DOI:** 10.3390/pharmaceutics14122774

**Published:** 2022-12-13

**Authors:** Yeong Chan Ryu, You-Rin Kim, Jiyeon Park, Sehee Choi, Won-Ji Ryu, Geon-Uk Kim, Eunhwan Kim, Yumi Hwang, Heejene Kim, Gyoonhee Han, Soung-Hoon Lee, Kang-Yell Choi

**Affiliations:** 1Department of Biotechnology, College of Life Science and Biotechnology, Yonsei University, Seoul 03722, Republic of Korea; 2CK Regeon Inc., B137 Engineering Research Park, 50 Yonsei-ro, Seodaemun-gu, Seoul 03722, Republic of Korea

**Keywords:** pyruvate kinase M2, metabolism, Wnt/β-catenin signaling pathway, hair follicle stem cell, hair regeneration

## Abstract

Hair follicle stem cells (HFSCs) utilize glycolytic metabolism during their activation and anagen induction. However, the role of pyruvate kinase M2 (PKM2), which catalyzes the final step of glycolysis, in hair regeneration has not been elucidated. In this study, we investigated the expression pattern and activity of PKM2 during the depilation-induced anagen progression in mice. We found that TEPP-46, a selective activator of PKM2, enhanced hair re-growth and proliferation of HFSCs. PKM2 expression was increased via up-regulation of Wnt/β-catenin signaling, which is involved in hair regeneration. Moreover, a combined treatment with KY19382, a small molecule that activates Wnt/β-catenin signaling, and TEPP-46 significantly enhanced hair re-growth and wound-induced hair follicle neogenesis (WIHN). These results indicate that simultaneous activation of the PKM2 and Wnt/β-catenin signaling could be a potential strategy for treating alopecia patients.

## 1. Introduction

A hair follicle is a mini-organ that regenerates itself through the hair cycle comprising three phases: growth phase (anagen), regression phase (catagen), and resting phase (telogen) [1]. Among various cell types that form the hair follicle, hair follicle stem cells (HFSCs) play the most important role in hair maintenance [2,3]. During the telogen-to-anagen transition, HFSCs in the bulge region receive hair growth-promoting signals from the dermal papilla cells [2]. Subsequently, HFSCs proliferate, and their progenies differentiate into various types of cells that organize the hair follicles [4,5]. It has recently been reported that HFSCs require aerobic glycolysis for metabolism [6,7]. In hair regeneration, keratin 15-positive HFSCs depend on the energy produced by glycolysis. On the other hand, differentiated proliferating cells mainly rely on oxidative phosphorylation, exhibiting a mature mitochondrial phenotype [8].

The pyruvate kinase (PK) is an enzyme that transfers phosphate from phosphoenolpyruvate to ADP and produces pyruvate and ATP in the last step of glycolysis [9]. PK acts as an enzyme that determines the rate of glycolysis [10]. There are four isoforms of PK in mammals: L, R, M1, and M2 [11]. Among them, pyruvate kinase M2 (PKM2) is mainly expressed in the regions where tissue regeneration occurs [12]. PKM2 plays an important role in proliferation in actively proliferating cells, such as cancer cells [13]. A recent study confirmed that PKM2 is induced during the telogen-to-anagen transition [14]. In addition, pyruvate, a product of glycolysis catalyzed by PK, is increased in HFSCs during the telogen-to-anagen transition [15]. However, the role of PKM2 in hair maintenance has not been identified yet.

The Wnt/β-catenin signaling pathway is a major pathway that regulates the proliferation and differentiation in many tissues [1,16,17]. In hair follicles, the canonical Wnt/β-catenin signaling pathway contributes to hair follicle morphogenesis, maintenance, and anagen induction [18,19,20]. The Wnt/β-catenin pathway is crucial in the activation, proliferation, and differentiation of HFSCs, particularly at the onset of the anagen phase [1]. In a previous study, PKM2 was identified as a direct transcriptional target of Wnt/β-catenin signaling in colorectal cancer [21]. However, a relationship between PKM2 and Wnt/β-catenin signaling in hair follicle regeneration has not been investigated.

The major available hair loss treatments, which have been approved by the FDA, are minoxidil and finasteride [22]. These drugs are effective for the growth of existing hair follicles; however, they are not effective in patients with severe hair loss because they do not induce hair regeneration [23]. Therefore, it is necessary to develop effective drugs for promoting hair regeneration. Although studies on hair growth via either activation of Wnt/β-catenin signaling or metabolism have been reported [14,15,24,25], no studies have confirmed the synergistic effect of enhancing both Wnt/β-catenin signaling and metabolism.

In this study, we identified that PKM2 plays a role in the activation of HFSCs and hair growth promotion. Pharmacological control of PKM2 activity modulated the proliferation of HFSCs and anagen induction. The expression level and localization of PKM2 correlated with those of β-catenin in the bulge region, where the stem cells reside in hair follicles. Moreover, simultaneous treatment with TEPP-46, a PKM2 activator [26], and KY19382, a small molecule that activates the Wnt/β-catenin signaling through the interference with the CXXC5 disheveled interaction, as well as the inhibition of GSK-3β [23,27], significantly promoted anagen induction and hair re-growth, followed by the activation of HFSCs. Furthermore, PKM2 expression was highly increased in the neogenic follicles during wound-induced hair neogenesis (WIHN), along with β-catenin and WIHN markers. This result indicated that PKM2 may play a role in the process of hair follicle neogenesis. Similarly, a combined treatment of KY19382 with TEPP-46 significantly enhanced the regeneration of hair follicles. Taken together, we found that both the increment of PKM2 levels by activating the Wnt/β-catenin signaling and the activation of PKM2 are important for hair regeneration. We suggest that dual activation of PKM2 and Wnt/β-catenin signaling could be a potential therapeutic strategy for alopecia patients.

## 2. Materials and Methods

### 2.1. Mouse Maintenance

All animal experiments had approval from the Institutional Animal Care and Use Committee of Yonsei University (IACUC-A-201906-926-01, IACUC-A-201907-929-01, IACUC-A-201903-880-03, IACUC-A-201905-910-03, IACUC-A-201905-906-03, IACUC-A-201908-940-01, IACUC-A-202009-1143-01). Animal studies were designated according to the ARRIVE guidelines [28].

Mice were housed in a computerized environmental managing system of micro-ventilation cages (Threeshine Inc., Seoul, Republic of Korea). Mice had free approach to food and water. Room temperature was managed at 24 °C, with a relative humidity of 40–70% and 12 light/12 dark cycles. The mice were raised with a normal diet and bedding from Central Lab Animal Inc. (Seoul, Republic of Korea). Mice were housed with no more than 5 animals per cage. The mice were randomly separated into each group. After the experiments were completed, the mice were euthanized using CO_2_ gas.

Considering the 3Rs (replacement, refinement, or reduction) of animal use, we first performed preliminary experiments to determine the effects of TEPP-46 and shikonin on hair growth through ex vivo vibrissa culture, and then we performed minimal animal experiments under optimal conditions.

### 2.2. Depilation-Induced Hair Cycle

Eight-week-old male C57BL/6N mice (Koatech Co., Namyangju-si, Gyeonggi-do, Republic of Korea) were anesthetized with 2,2,2-tribromoethanol (Sigma-Aldrich, St. Louis, MO, USA) through 400 mg/kg IP injections. The mouse backs were plucked to induce a synchronized hair cycle [29]. To monitor the protein expression pattern during the depilation-induced hair cycle, tissues were harvested on the indicated days after depilation, respectively. To confirm the inhibitory effect of shikonin on anagen induction, depilation was similarly performed in the dorsal skins of 8-week-old male C57BL/6N mice (Koatech Co.). One hundred and fifty μL of shikonin (Sigma-Aldrich) was applied topically daily for 5 or 11 days. Shikonin was dissolved in acetone (Duksan Pure Chemicals Co., Ansan-si, Gyeonggi-do, Republic of Korea). After the experiments were completed, the mice were euthanized using CO_2_ gas.

### 2.3. Hematoxylin and Eosin (H&E) Staining

Tissues were fixed overnight in 10% formalin, and paraffin was embedded. The paraffin-embedded skin tissues of 4 μm sections were attached to the slides. The slides were deparaffinized in three changes of xylene and rehydrated through a graded ethanol series. The sections were stained with hematoxylin for 5 min, followed by eosin for 1 min. The H&E-stained slides were photographed using a stereomicroscope (SMZ 745T; Nikon, Tokyo, Japan). The number of follicles was quantified by counting the hair follicles in H&E staining images. Dermal thickness was calculated using Image J software V1.48.

### 2.4. Immunohistochemistry (IHC)

The slides with skin tissues attached were deparaffinized and rehydrated. Antigens were retrieved by microwaving them for 15 min in 10 mM sodium citrate buffer. Antigens were blocked with 5% BSA in PBS and placed with the primary antibodies: rabbit anti-PKM2 (Cell Signaling Technology Cat# 4053, RRID: AB_10828320, 1:800, Beverly, MA, USA), rabbit anti-β-catenin (Abcam Cat# 16051, RRID: AB_443301, 1:100, Cambridge, MA, USA) or mouse anti-β-catenin (BD Transduction Laboratory Cat# 610154, RRID: AB_397555; 1:100, Lexington, MA, USA), rabbit anti-Ki67 (Abcam Cat# ab15580, RRID: AB_443209, 1:500), mouse anti-keratin 15 (Abcam Cat# ab80522, RRID: AB_1603675, 1:200), mouse anti-PCNA (Santa Cruz Technology Cat# sc-56, RRID: AB_628110; 1:500, Dallas, TX, USA), and rabbit anti-cytokeratin 17 (Abcam Cat# ab53707, RRID: AB_869865, 1:400) or rabbit anti-fibroblast growth factor (fgf) 9 (Abcam Cat# ab71395, RRID: AB_2103075, 1:200). After washing with PBS, antigens were placed with secondary antibodies: anti-mouse Alexa Fluor 488 (Life Technologies Cat# A11008, RRID: AB_143165, 1:500, Camarillo, CA, USA) or anti-rabbit Alexa Fluor 555 (Life Technologies Cat# A21428, RRID: AB_141784, 1:500). After washing with PBS, slides were counterstained with 4′,6-diamidino-2-phenylindole (DAPI; Sigma Aldrich). Fluorescent images were taken using a confocal microscope (LSM510; Carl Zeiss Inc., Germany).

### 2.5. Reverse Transcription and Quantitative Real-Time PCR

The total RNA was separated using Trizol reagent (Invitrogen, Carlsbad, CA, USA) according to the manufacturer’s descriptions. RNA (2 μg) was reverse transcribed using reverse transcriptases (Invitrogen) at 42 °C for 1 h. For real-time PCR, the resulting cDNA (1 μL) was amplified in a reaction mixture including 10 pmol of the primer set (Bioneer, Daejeon-si, Chungcheong-do, Republic of Korea) and iQ SYBR Green Supermix (QIAGEN, Hilden, Germany). The primer sequences were: *PKM2*, forward 5′-TGG CGC CCA TTA CCA GCG ACC-3′ and reverse 5′-TCC CTT CTT GAA GAA GCC TCG GGC CTT-3′ and *GAPDH*, forward 5′-ACC CAG AAG ACT GTG GAT GG-3′ and reverse 5′-GGA TGC AGG GAT GAT GTT CT-3′.

### 2.6. Pyruvate Kinase Activity Assay

Pyruvate kinase activity assay was previously described [30,31]. The tissues were lysed in RIPA buffer (150 mM NaCl; 10 mM Tris, pH 7.2; 0.1% SDS; 1% Triton X-100; 1% sodium deoxycholate; 5 mM EDTA). The lysates were centrifuged at 15,920× *g* at 4°C for 30 min. Equal amounts of protein (1 μg) were incubated with 100 mM KCl (Duksan Pure Chemicals Co.), 50 mM Tris (Amresco, Cleveland, OH) pH 7.5, 5 mM, 0.6 mM ADP, 0.5 mM phosphoenolpyruvate (Sigma-Aldrich), 10 μM fructose-1,6-bisphosphate (Sigma-Aldrich), 240 μM NADH (Sigma-Aldrich), and 8 units of lactate dehydrogenase (Sigma-Aldrich). The absorbance change of NADH’s oxidation was measured at 320 nm using a FLUOstar OPTIMA luminometer (BMG Labtech, Offenburg, Germany).

### 2.7. In Vivo Hair Re-Growth Test

Six-week-old male C57BL/6N mice were adapted to their new environment for 1 week. After 1 week of acclimation, seven-week-old mice, whose hair follicles entered the telogen phase [32], were anesthetized with 2,2,2-tribromoethanol (Sigma-Aldrich) through 400 mg/kg IP injections. After shaving mouse backs with a hair clipper, 150 μL of each drug was applied daily for 14, 28, or 35 days to the shaved dorsal skin of the mice; TEPP-46 (Cayman Chemical, Ann Arbor, MI, USA) was dissolved in 90% acetone (Duksan Pure Chemicals Co.) and 10% Tween 80 (Tokyo Chemical Industry Co., Tokyo, Japan); minoxidil (MNX; Tokyo Chemical Industry Co.) was dissolved in 30% (*v*/*v*) water, 20% propylene glycol (Junsei Chemical Industry Co., Tokyo, Japan), and 50% ethanol (Duksan Pure Chemicals Co., Gyeonggi-do, Republic of Korea); KY19382 was dissolved in polyethylene glycol 400 (PEG 400) (Sigma-Aldrich). To evaluate re-grown hair, re-grown hairs were collected using a clipper and were measured using a precision balance. After the experiments were completed, the mice were euthanized using CO_2_ gas.

### 2.8. Wound-Induced Hair Follicle Neogenesis Assay

Three-week-old male C57BL/6N mice were acclimated to their new environment for 3 days. After anesthesia with 2,2,2-tribromoethanol (400 mg/kg, IP), 1 cm^2^ full-thickness wounds were made on the backs of the mice under aseptic conditions. Twenty μL of each drug used in the hair regrowth tests was treated to the wound regions daily for up to 14 or 25 days. After the experiments were completed, the mice were euthanized using CO_2_ gas.

### 2.9. Whole-Mount Alkaline Phosphatase (ALP) Staining

The tissues were put into 5 mM EDTA in PBS at 37 °C for 6 h. The dermis layer was isolated under the microscope (Nikon) and incubated in 4% paraformaldehyde for 10 min and NBT/BCIP solution (NBT/BCIP tablets, Roche Diagnostics, Rotkreuz, Switzerland) for 10 min. The ALP staining images were obtained by using a stereomicroscope (Nikon). The number of neogenic follicles was quantified by counting dark blue dots.

### 2.10. Database

The gene expression profile results were obtained in NCBI’s GEO database (http://www.ncbi.nlm.nih.gov/geo/ (accessed on 1 December 2021)) using GEO accession number GSE45512.

### 2.11. Mouse Vibrissa Ex Vivo Culture

Mouse vibrissa follicles were extracted from 6-week-old C57BL/6N female mice under a stereomicroscope (Nikon). The separated follicles were cultured with 500 μL DMEM supplemented with 12.5 μg/mL gentamicin (Gibco, Gaithersburg, MD, USA) and 1% penicillin/streptomycin (Gibco) within a 24-well plate. The follicles were incubated with 0.1% (*v*/*v*) dimethyl sulfoxide (DMSO; Sigma-Aldrich), shikonin, or TEPP-46 for each well. All drugs were dissolved in DMSO. The culture medium was replaced every 2 days, and the hair shaft length of vibrissa follicles was measured at 10 days. The follicle was visualized with a stereomicroscope (Nikon). The length was measured using Image J software V1.48.

### 2.12. Statistical Analyses

All experiments were designed to establish blinding and randomization. All group sizes represent the numbers of experimental independent values, and these independent values were used to evaluate statistical analyses. Each experiment was performed at least three times, and the results were expressed as mean ± standard error of the mean. Statistical analyses for two-group data were performed using an unpaired, two-tailed Student’s t-test. For multi-group data, analyses were tested using one-way ANOVA, followed by Tukey’s test using Prism software V5.01 (Graphpad, San Diego, CA, USA). Statistical significance is indicated in the figures as follows: * *p* < 0.05, ** *p* < 0.005, *** *p*< 0.0005.

## 3. Results

### 3.1. PKM2 Increases in HFSCs during the Anagen Induction

We analyzed the expression profiles of PKM2 to confirm any differences in its expression in alopecia patients, compared to healthy controls (GEO: GSE45512). PKM2 expression levels were decreased in the alopecia patients, compared to the healthy controls (Appendix A). This result correlates with a previous study that showed an increase in the levels of PKM2 and pyruvate during the telogen-to-anagen transition [14]. To further characterize the role of PKM2 in the activation of stem cells during anagen induction, we assessed the expression pattern of PKM2 during the depilation-induced hair cycle in mice [29] (Figure 1a). The levels of β-catenin, essential for anagen induction [33] and PKM2, were similarly increased in a time-dependent manner during the telogen-to-anagen transition (Figure 1b–e). The expression level of Ki67, a proliferation marker, was significantly increased in the region where keratin 15-positive stem cells were present [34] (Figure 1b). Similarly, PKM2 mRNA levels were elevated during anagen induction (Figure 1f). Moreover, PK activity was elevated during anagen induction (Figure 1g). These results indicate that both the expression and activity of PKM2 were increased during anagen induction and HFSC activation.

### 3.2. Activation of PKM2 Induces HFSC Proliferation and Hair Growth

To investigate the effect of TEPP-46, a selective PKM2 activator [26], on hair re-growth, we utilized the ex vivo vibrissa follicle system. The vibrissa hair shaft length was increased by TEPP-46 treatment in a dose-dependent manner (Appendix A). To further confirm the effect of PKM2 activation on hair re-growth in vivo, we administered TEPP-46 or minoxidil (MNX), a Food and Drug Administration (FDA)-approved alopecia drug as a positive control [35], on the shaved area of mouse dorsal skins. The hair follicles treated with TEPP-46 or MNX entered the anagen phase, while those treated with the vehicles remained in the telogen phase, even after 35 days (Figure 2a). The maximal hair re-growth effect of TEPP-46 was observed at 0.5 mM (Appendix A), and the hair promoting effect over MNX was confirmed by H&E staining (Figure 2b). There was an increase in the relative number of hair follicles and dermal thickness in the TEPP-46-treated group, compared to those in the vehicle- or MNX-treated groups (Figure 2c,d). IHC analyses showed that the proliferation markers, Ki67 and PCNA, were significantly increased in HFSCs of the TEPP-46-treated group, compared to those of the vehicle- or MNX-treated groups (Figure 2e and Appendix A). The specific increase in PK activity was confirmed in the TEPP-46-treated group (Figure 2f).

To further characterize the effect of PKM2 activity on hair growth, we tested the effects of shikonin, a selective PKM2 inhibitor [36]. The vibrissa hair shaft length was decreased by shikonin treatment in a dose-dependent manner (Appendix A). In depilation-induced mice, anagen induction was delayed in the shikonin-treated group but not in the vehicle-treated control group that entered the anagen phase (Figure 2g). H&E staining revealed that the number of hair follicles and dermal thickness in the shikonin-treated group were reduced, compared to those in the vehicle-treated group (Figure 2h–j). The proliferation of stem cells in the shikonin-treated group was significantly reduced, as revealed by the IHC analyses (Figure 2k and Appendix A). PK activity was also reduced in the shikonin-treated group, compared to that in the control group (Figure 2l). In summary, the proliferation of HFSCs depends on PK activity during hair re-growth.

### 3.3. PKM2 Expression Depends on Wnt/β-Catenin Signaling

To determine the correlation between PKM2 expression and Wnt/β-catenin signaling, we analyzed the PKM2 and β-catenin during the depilation-induced anagen progression. In the bulge region, the expression pattern of PKM2 was similar to that of β-catenin and PCNA (Appendix A). Quantitative analyses revealed that the levels of PKM2 and nuclear β-catenin were increased simultaneously during the telogen-to-anagen transition (Appendix A).

In a recent study, we noted that PKM2 expression was induced by activating the Wnt/β-catenin signaling in colorectal cancer [21]. To investigate the effect of Wnt/β-catenin signaling on the PKM2 expression in hair follicles, we assessed the effects of KY19382, a small molecular Wnt/β-catenin signaling activator that stimulates hair growth and HFSC activation [23]. The expressions of β-catenin and PKM2 were increased only in the KY19382-treated group in the stem cell regions (Appendix A). Overall, PKM2 expression correlated with the Wnt/β-catenin signaling in HFSC activation.

### 3.4. Combined Treatment with Activators of PKM2 and Wnt/β-Catenin Signaling Promotes Hair Re-Growth

To investigate the combined effect of the activators of PKM2 and Wnt/β-catenin signaling on hair re-growth, TEPP-46 was co-treated with KY19382 on dorsal skins of mice. Hair re-growth was significantly enhanced by combined treatment with TEPP-46 and KY19382, as shown by the measurement of hair weight (Figure 3a,b). Moreover, anagen induction, relative number of hair follicles, and dermal thickness were all significantly increased by combined treatment (Figure 3c–e). IHC staining showed that both β-catenin and PKM2 were increased in the HFSC regions only in the KY19382-treated groups (Figure 3f–i). The proliferation markers were specifically increased in stem cells after the combined treatment (Figure 3c and Appendix A). PK activity was increased only in the TEPP-46-treated groups (Figure 3j). In summary, hair re-growth was enhanced by the activation of both the Wnt/β-catenin signaling and PKM2.

### 3.5. PKM2 Is Highly Expressed in the Neogenic Follicles during WIHN

The WIHN assay was used to assess hair regeneration (Appendix A) and was induced via the Wnt/β-catenin signaling [37]. The PKM2 was highly increased together with β-catenin in neogenic hair follicles at post-wounding days (PWDs) 17 and 25 (Figure 4b). As monitored by fgf 9 and keratin 17, the markers for WIHN [38,39], the formation of new hair follicles correlated with the elevation of β-catenin and PKM2 during the WIHN process (Appendix A). Correspondingly, the *PKM2* mRNA level, as well as PK activity, were increased at PWDs 17 and 25 (Appendix A). Overall, PKM2 was highly expressed in the neogenic follicles with the induction of β-catenin and WIHN markers, and PK activity was also increased during the formation of new hair follicles. Therefore, PKM2 may contribute to the formation of neogenic hair follicles.

### 3.6. Combined Treatment with TEPP-46 and KY19382 Enhances WIHN

To determine the combined effect of KY19382 and TEPP-46 on WIHN, KY19382 was co-treated with TEPP-46 on mice wounds. To confirm the fully regenerative follicles, we conducted whole-mount ALP staining using mouse wounds at 25 days after drug application. Whole-mount ALP staining showed that TEPP-46 induced WIHN. The combined treatment with KY19382 and TEPP-46 further increased the number of neogenic hair follicles (Figure 4a,b). To observe the initial formation of neogenic follicles, we performed H&E and IHC staining using mouse tissue at 14 days after drug treatment. The increment of neogenic hair follicles by treatment with TEPP-46 and/or KY19382 was further confirmed by H&E staining (Figure 4c). IHC analyses revealed that both β-catenin and PKM2 were increased only in the KY19382-treated groups (Figure 4d–g). The combined treatment increased proliferation of epidermal and dermal cells with the increment of the WIHN markers (Figure 4d and Appendix A). Moreover, PK activity was elevated only in the TEPP-46-treated groups (Figure 4h). These results showed that PKM2 was increased by the up-regulation of the Wnt/β-catenin signaling in neogenic follicles. The formation of neogenic follicles was significantly promoted by the combined treatment with TEPP-46 and KY19382.

## 4. Discussion

Alopecia is characterized by the shortening of the anagen phase and miniaturization of hair follicles [40]. The number of alopecia patients is rapidly increasing, regardless of sex and age [41,42]. Hair loss is mainly caused by a decrease in HFSC activity and regenerative capability of hair follicles [43]. For this reason, many studies aiming to activate the stem cell and regenerative capacity of hair follicles are currently increasing [14,15]. A recent study reported that active glycolysis occurs when the HFSCs are activated, whereas anagen induction is delayed if adequate glycolysis is not performed [14]. Previous studies also suggested that the up-regulation of glycolysis can be an effective strategy for anagen induction and hair growth promotion [14,15]. In addition, activation of stem cells using small molecules inducing the Wnt/β-catenin signaling pathway was suggested as a strategy for hair re-growth and regeneration [23,44]. Therefore, small molecular control of HFSC metabolism is a promising strategy for hair regeneration.

PKM2 plays a crucial role in aerobic glycolysis by catalyzing the last rate-limiting step of glycolysis [9,45]. The PKM2 is highly expressed in stem cells, proliferating cells, and embryonic tissues that retain high anabolic activity [46]. The PKM2 is essential in inhibiting oxidative phosphorylation activity and inducing the proliferation of stem cells, including hematopoietic stem cells [47]. A recent study also showed that PKM2 and pyruvate increased in HFSCs during the telogen-to-anagen transition of the hair cycle [14]. In addition, in this study, we found that pharmacological activation of PKM2 significantly induced HFSC proliferation. These studies suggest that PKM2 could be a potential target to induce hair regeneration via HFSC activation.

In this study, we investigated the roles of activity and expression levels of PKM2 in hair regeneration. The control of PKM2 activity by the treatment of TEPP-46 or shikonin affects hair growth both ex vivo and in vivo. The role of PKM2 level in hair promotion is especially related to Wnt/β-catenin signaling, as evidenced by the effect of KY19382 in hair growth and the correlation between expression levels of β-catenin and PKM2. The pathophysiological role of PKM2 level in hair growth is supported by reduced PKM2 levels in alopecia patients. Furthermore, we found that combined treatment with Wnt/β-catenin signaling and PKM2 activators further enhanced hair re-growth and WIHN, compared to single treatment. This further hair growth is attributed to the fact that the combined treatment activates and increases PKM2 through Wnt/β-catenin signaling. These data indicate that both the level and activity of PKM2 play a role in hair growth. Moreover, an approach using both PKM2 and Wnt/β-catenin signaling activators could be an effective strategy for hair regeneration.

## Figures and Tables

**Figure 1 pharmaceutics-14-02774-f001:**
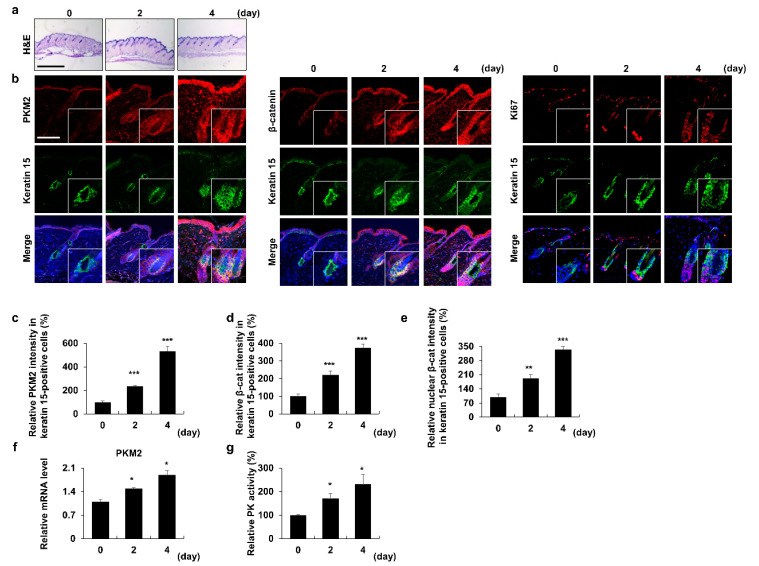
Expression profiles of PKM2 in the stem cell region of hair follicles during the telogen-to-anagen transition. The dorsal skins of 8-week-old C57BL/6N male mice were depilated and harvested on the indicated days, as described in the Materials and Methods section. (**a**) H&E staining of mouse dorsal skins. (**b**) IHC staining for keratin 15, PKM2, β-catenin, and Ki67. Magnified images indicate the keratin 15-positive stem cell region. (**c**–**e**) Quantitative analyses of the intensity of the fluorescence signals for PKM2 and β-catenin in the keratin 15-positive region (*n* = 15). Relative mRNA level of *Pkm2* (**f**) and quantitative measurements of PK activity (**g**) in mouse dorsal skins (*n* = 3). Scale bars = 100 µm. Values are expressed as mean ± SEM. Student’s *t*-test (* *p* < 0.05, ** *p* < 0.005, *** *p* < 0.0005).

**Figure 2 pharmaceutics-14-02774-f002:**
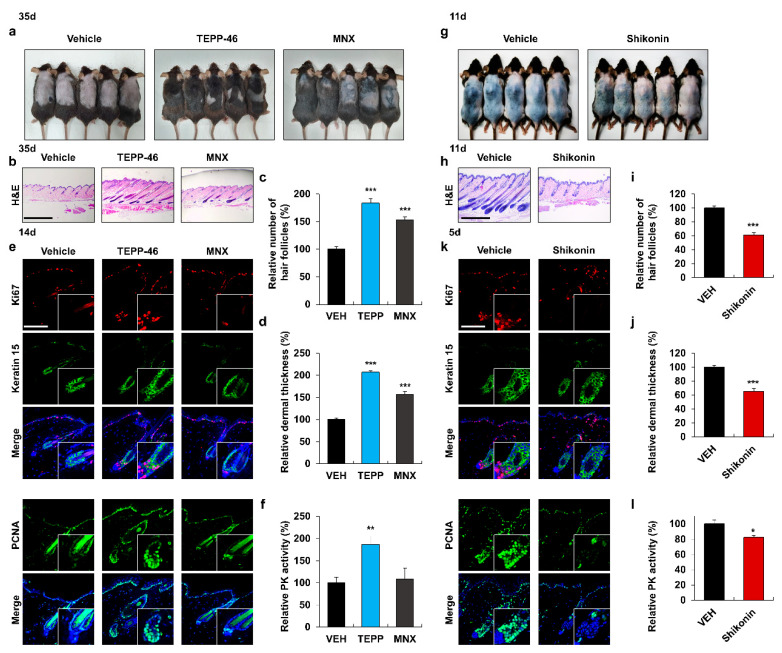
Effects of PKM2 activator TEPP-46 on hair re-growth and HFSC proliferation. (**a**–**f**) The dorsal skins of 7-week-old C57BL/6N male mice were shaved and topically treated daily with vehicles, TEPP-46 (0.5 mM), or MNX (150 mM) for 14 or 35 days. (**a**) Gross images of regrown hair in mice treated for 35 days. (**b**) H&E staining of the dorsal skins treated with TEPP-46 or MNX for 35 days. (**c**) Quantitative evaluation of the number of hair follicles and (**d**) dermal thickness in H&E-stained skin tissues (*n* = 15). (**e**) IHC analyses for keratin 15, Ki67, and PCNA in hair follicles of mice skins treated for 14 days. Magnified images show the stem cell region. (**f**) PK activity of dorsal skins treated for 14 days (*n* = 4). (**g**–**l**) The dorsal skins of 8-week-old C57BL/6N male mice were depilated and topically applied with vehicle or shikonin (0.1 mM) daily for 5 or 11 days. (**g**) Gross images and (**h**) H&E staining of mice treated with shikonin for 11 days. (**i**,**j**) Quantitative measurements of the number of hair follicles and dermal thickness in H&E staining (*n* = 15). (**k**) IHC staining for Ki67, keratin 15, and PCNA of hair follicles in mice skins treated for 5 days. Enlarged pictures display the stem cell region. (**l**) PK activity of skin tissues treated for 5 days (*n* = 3). Scale bars = 100 µm. Values are expressed as mean ± SEM. Student’s *t*-test (* *p* < 0.05, ** *p* < 0.005, *** *p* < 0.0005).

**Figure 3 pharmaceutics-14-02774-f003:**
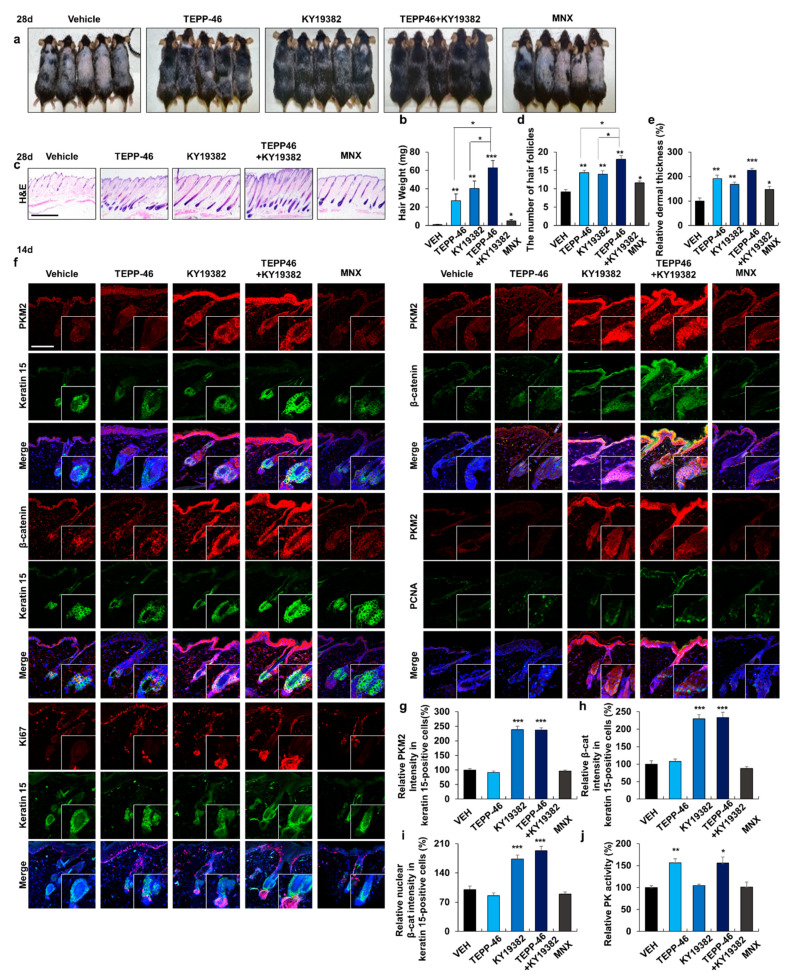
Effects of combined treatment with activators of PKM2 and Wnt/β-catenin signaling on HFSC activation and hair re-growth. The dorsal skins of male mice were shaved and topically treated daily with vehicles, TEPP-46 (0.5 mM), KY19382 (0.5 mM), or MNX (150 mM) for 14 or 28 days. (**a**) Gross images of hair re-growth in mice treated with compounds for 28 days. (**b**) The measurements of regrown hair weight (*n* = 5). (**c**) H&E staining of dorsal skins treated for 28 days. (**d**,**e**) Quantitative estimations of dermal thickness and the number of hair follicles in H&E-stained images (*n* = 5). (**f**) IHC staining for PKM2, β-catenin, Ki67, keratin 15, and PCNA in skins treated for 14 days. Magnified images display the keratin 15-positive stem cell region. (**g**) Quantitative measurements of the intensity of the fluorescence signals for PKM2 in keratin 15-positive stem cell region (*n* = 15). (**h**,**i**) Quantification of the intensity of the fluorescence signals for total or nuclear β-catenin in the stem cell region (*n* = 15). (**j**) PK activity of skins treated for 14 days (*n* = 3). Scale bars = 100 µm. Values are expressed as mean ± SEM. Student’s *t*-test (* *p* < 0.05, ** *p* < 0.005, *** *p* < 0.0005).

**Figure 4 pharmaceutics-14-02774-f004:**
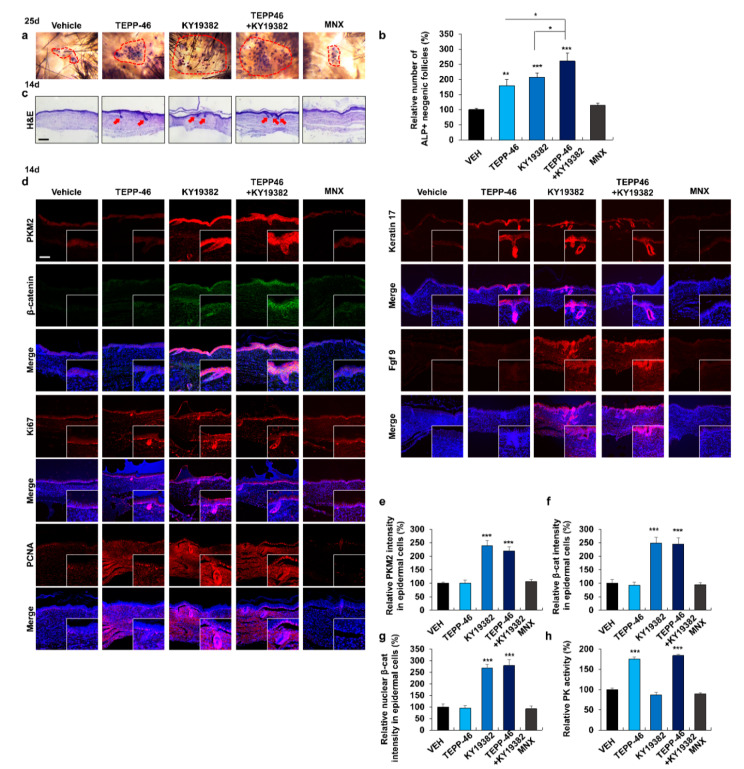
Influences of combined treatment with KY19382 and TEPP-46 on WIHN. The dorsal skins of 3-week-old C57BL/6N male mice were wounded and applied daily with vehicles, KY19382 (0.5 mM), TEPP-46 (0.5 mM), or MNX (150 mM) for 14 or 25 days. (**a**) Whole-mount ALP staining and (**b**) quantification of ALP-positive neogenic follicles at day 25 (*n* = 8). Inside area of dashed lines indicate ALP-positive neogenic follicles. (**c**) H&E staining of wounded skins treated for 14 days. Arrows indicate the neogenic follicles. (**d**) IHC staining for PKM2, β-catenin, Ki67, PCNA, keratin 17, and fgf 9 in wounded skins treated for 14 days. Magnified images show the neogenic follicles. (**e**–**g**) Quantification of the intensity of the fluorescence signals for PKM2 and β-catenin (*n* = 15). (**h**) PK activity of wounded skins treated for 14 days (*n* = 4). Scale bars = 100 µm. Values are expressed as mean ± SEM. Student’s *t*-test (* *p* < 0.05, ** *p* < 0.005, *** *p* < 0.0005).

## Data Availability

The data that support the findings of this study are available upon request from the corresponding author. The data are not publicly available due to privacy or ethical restrictions.

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
