# Peer review of "Pyruvate Kinase M2 Promotes Hair Regeneration by Connecting Metabolic and Wnt/β-Catenin Signaling"

_pharmaceutics, 2022, doi:10.3390/pharmaceutics14122774_

Round 1

Reviewer 1 Report

In the work “Pyruvate kinase M2 promotes hair regeneration by connecting metabolic and Wnt/β-catenin signaling”, the authors studied the role of pyruvate kinase M2 (PKM2) in hair regeneration.  The manuscript is well presented, with a good level of English. Some data is missing. Also, some questions should be addressed before acceptance:

·      Introduction section – Authors should compare their strategy with others available for the treatment of alopecia. Please add more information.

·        Page 3, line 109 – Please add more information regarding the immunohistochemistry assay.

·    Page 3, line 143 – Authors refer that “Seven-week-old mice, whose entered the telogen phase”. Please indicate how do the authors had a complete control over the hair growth stage.

·        Page 4, line 145 – 150 – Is not clear how the treatments are applied? Same area? Different areas? Include why minoxidil is being included in the study.  Please avoid sentences such “same as those used in the previous study.”. Authors should include all the information necessary to reply the work.

·        Page 5, line 192 – authors refer to a supplementary figure 1a however any figure can be found in Supplementary material. Please revise.

·        Page 6, line 216 – supplementary figures 1b and 1c cannot be found on supplementary material

·        Page 6, lines 217-218 – indicate where the TEP-46 and the minoxidil were administered

·        Page 6, line 222, 227 and 232 - supplementary figures 1d, 1e, 1f and 1g cannot be found on supplementary material

·        Page 7, Figure 2 – increase panels A and G for better visualization

·        Page 7, lines 261 and 263 – supplementary 2a, 2b, 2c, 2d and 3c cannot be found on supplementary material

·        Page 9, Figure 3 – Please increase panel A for better visualization. How do authors explain the lower activity of minoxidil when compared with TEP-46 treatment, KY19382 treatment or with the combination of these? Hair weight protocol should be included in materials and methods section.

·        Page 10, Line 298 – Supplementary figure 4a cannot be found on supplementary material

·        Page 11, Figure 4 – In panel A and B the authors refer to day 25, however for panel B appears day 14 in the figure.  Why did the authors present the other results for day 14 and not for day 25? Please explain in better detail. Moreover, on page 9, line 300 authors refer to day 17 and no results can be found for day 17.   

·        References section seem to be written in a different text font.

Author Response

* RESPONSES TO REVIEWERS

Reviewer 1)

  1. Reviewer A stated that the authors studied the role of pyruvate kinase M2 (PKM2) in hair regeneration.  The manuscript is well presented, with a good level of He or she also mentioned that some data were missing. Also, some questions should be addressed before acceptance.

The authors thank to the kind comments from Reviewer 1. To further improve the quality of the manuscript, we carefully revised the manuscript by addressing the following issues suggested by Reviewer 1.

  • Introduction section – Authors should compare their strategy with others available for the treatment of alopecia. Please add more information.

The major available hair loss treatments which have been approved by FDA are minoxidil and finasteride. These drugs are effective for the growth of existing hair follicles, however they are not effective in patients with severe hair loss because they do not induce hair regeneration. Therefore, it is necessary to develop effective drugs for promoting hair regeneration. Although studies on hair growth via either activation of Wnt/β-catenin signaling or metabolism have been reported, no studies have confirmed the synergistic effect of enhancing both Wnt/β-catenin signaling and metabolism. We have described this in the revised manuscript (Page 5, lines 6 – 12; Introduction section in the revised manuscript).

  • Page 3, line 109 – Please add more information regarding the immunohistochemistry assay.

We described the detailed method for IHC experiment in the Materials and Methods section (Page 8, lines 1 – 17; Method section 2.4. in the revised manuscript).

  • Page 3, line 143 – Authors refer that “Seven-week-old mice, whose entered the telogen phase”. Please indicate how do the authors had a complete control over the hair growth stage.

It is known that the refractory telogen starts at 7 weeks in wild-type C57BL/6N mouse [1]. This was referred in the revised manuscript (Page 9, line 18; Method section 2.7. in the revised manuscript).

  • Page 4, line 145 – 150 – Is not clear how the treatments are applied? Same area? Different areas? Include why minoxidil is being included in the study. Please avoid sentences such “same as those used in the previous study.”. Authors should include all the information necessary to reply the work.

According to the reviewer’s request, we included all the information necessary for this work (Page 10, lines 1 – 3; Method section 2.7. in the revised manuscript) rather than indirectly explaining it by citing references.

We added information on which area of ​​the mouse was treated with the drug in the revised manuscript. Each drug was daily applied to the shaved dorsal skin of mice (Page 9, line 21; Method section 2.7. in the revised manuscript). We have already mentioned in the manuscript that minoxidil was used as a positive control because it is an FDA-approved topical hair loss treatment (Page 13, lines 5 – 6; Result section 3.2. in the revised manuscript).

  • Page 5, line 192 – authors refer to a supplementary figure 1a however any figures cannot be found in Supplementary material. Please revise.

We think there might be an error while uploading the files of supplementary figures. We will try to upload them properly again.

  • Page 6, line 216 – supplementary figures 1b and 1c cannot be found on supplementary material

It seems that there was an error in the process of uploading the supplementary figures, so we will upload it again.

  • Page 6, lines 217-218 – indicate where the TEP-46 and the minoxidil were administered

We described that drugs were applied to the shaved area of mouse dorsal skins in the revised manuscript (Page 13, line 6; Result section 3.2. in the revised manuscript).

  • Page 6, line 222, 227 and 232 - supplementary figures 1d, 1e, 1f and 1g cannot be found on supplementary material

We will re-upload the supplementary figures which may not have been uploaded due to an error.

  • Page 7, Figure 2 – increase panels A and G for better visualization

We changed Figure 2A and 2G with the magnified ones.

  • Page 7, lines 261 and 263 – supplementary 2a, 2b, 2c, 2d and 3c cannot be found on supplementary material

We will re-post supplementary figures.

  • Page 9, Figure 3 – Please increase panel A for better visualization. How do authors explain the lower activity of minoxidil when compared with TEP-46 treatment, KY19382 treatment or with the combination of these? Hair weight protocol should be included in materials and methods section.

We increased the size of Figure 3A as requested by the reviewer.

The reason for the different activity of minoxidil is that the duration of drug treatment is different. The effect of hair follicle regrowth was confirmed following drug treatment for 35 days in the TEPP-46 treatment experiment, but it was confirmed following drug treatment for 28 days in the KY19382 treatment experiment. The descriptions of drug treatment have already been described in the manuscript.

One of the methods to accurately evaluate hair re-growth is to collect the re-grown hairs by using a hair clipper and weigh them by using a precision balance [2-4]. We newly described this in the Materials and methods section of the revised manuscript (Page 10, lines 4 – 5; Method section 2.7. in the revised manuscript).

  • Page 10, Line 298 – Supplementary figure 4a cannot be found on supplementary material

We will re-upload supplementary figures.

  • Page 11, Figure 4 – In panel A and B the authors refer to day 25, however for panel B appears day 14 in the figure. Why did the authors present the other results for day 14 and not for day 25? Please explain in better detail. Moreover, on page 9, line 300 authors refer to day 17 and no results can be found for day 17.

The Figures 4A and 4B, referred to as day 25, are whole mount staining of the wound on day 25 and its quantitative data. Perhaps the data that the reviewer confused as being from day 14 are the H&E-stained images in Figure 4C.

We also described in detail the purpose of H&E and whole-mount staining performed on days 14 and 25, respectively, in the revised manuscript (Page 16, lines 8-9; Page 16, lines 4-5; Result Section 3.6. in the revised manuscript). Based on other literatures [4,5], H&E staining was performed on day 14 to show the initial formation of neogenic follicles, whereas whole mount staining was performed on day 25 to accurately quantify and analyze fully regenerated follicles.

The day 17 described on page 9, line 300 is for an explanation of the supplementary figures, but it seems that there was an error in the process of uploading the supplementary data. We will try to upload properly again.

  • References section seems to be written in a different text font.

We matched the text font in the references section as the manuscript text font.

References

  1. Lin, X.; Zhu, L.; He, J. Morphogenesis, Growth Cycle and Molecular Regulation of Hair Follicles. Frontiers in cell and developmental biology 2022, 10, 899095, doi:10.3389/fcell.2022.899095.
  2. Lee, S.H.; Yoon, J.; Shin, S.H.; Zahoor, M.; Kim, H.J.; Park, P.J.; Park, W.S.; Min do, S.; Kim, H.Y.; Choi, K.Y. Valproic acid induces hair regeneration in murine model and activates alkaline phosphatase activity in human dermal papilla cells. PloS one 2012, 7, e34152, doi:10.1371/journal.pone.0034152.
  3. Lee, S.H.; Seo, S.H.; Lee, D.H.; Pi, L.Q.; Lee, W.S.; Choi, K.Y. Targeting of CXXC5 by a Competing Peptide Stimulates Hair Regrowth and Wound-Induced Hair Neogenesis. The Journal of investigative dermatology 2017, 137, 2260-2269, doi:10.1016/j.jid.2017.04.038.
  4. Ryu, Y.C.; Lee, D.H.; Shim, J.; Park, J.; Kim, Y.R.; Choi, S.; Bak, S.S.; Sung, Y.K.; Lee, S.H.; Choi, K.Y. KY19382, a novel activator of Wnt/β-catenin signalling, promotes hair regrowth and hair follicle neogenesis. British journal of pharmacology 2021, 178, 2533-2546, doi:10.1111/bph.15438.
  5. Ito, M.; Yang, Z.; Andl, T.; Cui, C.; Kim, N.; Millar, S.E.; Cotsarelis, G. Wnt-dependent de novo hair follicle regeneration in adult mouse skin after wounding. Nature 2007, 447, 316-320, doi:10.1038/nature05766.
  6. Christofk, H.R.; Vander Heiden, M.G.; Harris, M.H.; Ramanathan, A.; Gerszten, R.E.; Wei, R.; Fleming, M.D.; Schreiber, S.L.; Cantley, L.C. The M2 splice isoform of pyruvate kinase is important for cancer metabolism and tumour growth. Nature 2008, 452, 230-233, doi:10.1038/nature06734.
  7. Christofk, H.R.; Vander Heiden, M.G.; Wu, N.; Asara, J.M.; Cantley, L.C. Pyruvate kinase M2 is a phosphotyrosine-binding protein. Nature 2008, 452, 181-186, doi:10.1038/nature06667.

Reviewer 2 Report

1.      What was the motivation for this study? The authors need to justify the same.

2.      Introduction section seems a bit haphazard and needs reorganization. Add the latest references about PKM2 adding a few parts in the introduction section. A recent study showed various important roles pf PKM2.  https://doi.org/10.3390/ijms232113172; https://doi.org/10.1016/B978-0-323-91287-7.00027-2

3.      Eight-week-old male C57BL/6N mice………All the mice used in the study belong to this?

4.      Pyruvate kinase activity assay. Cite a proper suitable reference.

5.      Six-week-old male C57BL/6N mice were adapted to their new environment for week. Seven-week-old mice, whose hair follicles entered the telogen phase, were anesthetized with 2,2,2-tribromoethanol (Sigma-Aldrich) through 400 mg/kg IP injection. Why age different?

6.      There are many long sentences in the manuscript that need to be restructured.

Author Response

* RESPONSES TO REVIEWERS

Reviewer 2)

  • What was the motivation for this study? The authors need to justify the motivation.

Thanks for the kind comments from the Reviewer 2. We addressed the issues below raised by the reviewer.

The clinically available, FDA-approved drugs, minoxidil and finasteride, promote hair regrowth, but not hair regeneration. Therefore, we tested effectiveness of the modulators of Wnt/β-catenin signaling and metabolism to compare their hair regeneration ability and ultimately to develop drugs inducing hair regeneration. In particular, based on the reports on the promoting effect of hair growth by metabolic regulation and the role of PKM2 in Wnt/β-catenin signaling activation, we conducted experiments to confirm synergistic effects of Wnt/β-catenin signaling activator and PKM2 activator on hair regeneration. We added this in the introduction section of the revised manuscript (Page 5, lines 6 – 12; Introduction section in the revised manuscript).

  • Introduction section seems a bit haphazard and needs reorganization. Add the latest references about PKM2 adding a few parts in the introduction section. A recent study showed various important roles pf PKM2. https://doi.org/10.3390/ijms232113172; https://doi.org/10.1016/B978-0-323-91287-7.00027-2

As the reviewer advised, we added and referenced the contents of the two recent studies in the revised manuscript (Page 4, line 14 and line 16 – 17; Introduction section in the revised manuscript).

  • Eight-week-old male C57BL/6N mice………All the mice used in the study belong to this?

The mice questioned by the reviewer were used for the depilation experiment (page 7, line 3; Method section 2.2. in the revised manuscript). We have already mentioned about mouse age for different in vivo experiments in each Materials and Methods section (Page 9, line 17 – 18; page 10, line 9 in the revised manuscript).

  • Pyruvate kinase activity assay. Cite a proper suitable reference.

Previously, we have already referred to a paper describing the pyruvate kinase activity assay [6]. We added another study with an experimental description [7] on the reviewer’s advice (Page 9, line 6; Method section 2.6. in the revised manuscript).

  • Six-week-old male C57BL/6N mice were adapted to their new environment for week. Seven-week-old mice, whose hair follicles entered the telogen phase, were anesthetized with 2,2,2-tribromoethanol (Sigma-Aldrich) through 400 mg/kg IP injection. Why age different?

When performing in vivo hair re-growth experiments, the purchased mouse underwent 1 week of acclimation. Therefore, the age is not different. After acclimating 6-week-old mice for 1 week, 7-week-old mice were used in the experiments. To avoid confusion, the sentence ‘After 1 week acclimation’ was added in the revised manuscript (Page 9, line 17 – 18; Method section 2.7. in the revised manuscript).

  • There are many long sentences in the manuscript that need to be restructured.

As instructed by the reviewer, we divided the long sentences into two sentences to make it easier to read (Page 4, line 9 – 11, Introduction section; page 16, line 6 – 7, Result section 3.6.; page 17, line 24 – page 18, line 3, Discussion section; page 18 line 3 – 5, Discussion section in the revised manuscript).

References

  1. Lin, X.; Zhu, L.; He, J. Morphogenesis, Growth Cycle and Molecular Regulation of Hair Follicles. Frontiers in cell and developmental biology 2022, 10, 899095, doi:10.3389/fcell.2022.899095.
  2. Lee, S.H.; Yoon, J.; Shin, S.H.; Zahoor, M.; Kim, H.J.; Park, P.J.; Park, W.S.; Min do, S.; Kim, H.Y.; Choi, K.Y. Valproic acid induces hair regeneration in murine model and activates alkaline phosphatase activity in human dermal papilla cells. PloS one 2012, 7, e34152, doi:10.1371/journal.pone.0034152.
  3. Lee, S.H.; Seo, S.H.; Lee, D.H.; Pi, L.Q.; Lee, W.S.; Choi, K.Y. Targeting of CXXC5 by a Competing Peptide Stimulates Hair Regrowth and Wound-Induced Hair Neogenesis. The Journal of investigative dermatology 2017, 137, 2260-2269, doi:10.1016/j.jid.2017.04.038.
  4. Ryu, Y.C.; Lee, D.H.; Shim, J.; Park, J.; Kim, Y.R.; Choi, S.; Bak, S.S.; Sung, Y.K.; Lee, S.H.; Choi, K.Y. KY19382, a novel activator of Wnt/β-catenin signalling, promotes hair regrowth and hair follicle neogenesis. British journal of pharmacology 2021, 178, 2533-2546, doi:10.1111/bph.15438.
  5. Ito, M.; Yang, Z.; Andl, T.; Cui, C.; Kim, N.; Millar, S.E.; Cotsarelis, G. Wnt-dependent de novo hair follicle regeneration in adult mouse skin after wounding. Nature 2007, 447, 316-320, doi:10.1038/nature05766.
  6. Christofk, H.R.; Vander Heiden, M.G.; Harris, M.H.; Ramanathan, A.; Gerszten, R.E.; Wei, R.; Fleming, M.D.; Schreiber, S.L.; Cantley, L.C. The M2 splice isoform of pyruvate kinase is important for cancer metabolism and tumour growth. Nature 2008, 452, 230-233, doi:10.1038/nature06734.
  7. Christofk, H.R.; Vander Heiden, M.G.; Wu, N.; Asara, J.M.; Cantley, L.C. Pyruvate kinase M2 is a phosphotyrosine-binding protein. Nature 2008, 452, 181-186, doi:10.1038/nature06667.

Round 2

Reviewer 1 Report

The authors addressed the previous comments.

However no Figures and Supplementary Figures can be found on the revised version of the manuscript. 

Please revise.